# The Role of Cognitive Performance and Physical Functions in the Association between Age and Gait Speed: A Mediation Study

**DOI:** 10.3390/geriatrics7040073

**Published:** 2022-07-07

**Authors:** Marcelo de Maio Nascimento, Élvio Rúbio Gouveia, Bruna R. Gouveia, Adilson Marques, Priscila Marconcin, Cíntia França, Andreas Ihle

**Affiliations:** 1Department of Physical Education, Federal University of Vale do São Francisco, Petrolina 56304-917, Brazil; 2Department of Physical Education and Sport, University of Madeira, 9020-105 Funchal, Portugal; erubiog@staff.uma.pt (É.R.G.); cintia.franca@staff.uma.pt (C.F.); 3LARSYS-Laboratory for Robotics and Engineering System, Interactive Technologies Institute, 9020-105 Funchal, Portugal; bgouveia@esesjcluny.pt; 4Center for the Interdisciplinary Study of Gerontology and Vulnerability, University of Geneva, 1205 Geneva, Switzerland; andreas.ihle@unige.ch; 5Regional Directorate of Health, Secretary of Health of the Autonomous Region of Madeira, 9004-515 Funchal, Portugal; 6Saint Joseph of Cluny Higher School of Nursing, 9050-535 Funchal, Portugal; 7CIPER-Interdisciplinary Centre for the Study of Human Performance Faculty of Human Kinetics, University of Lisbon, 1495-751 Lisbon, Portugal; amarques@fmh.ulisboa.pt (A.M.); priscilamarconcin@fmh.ulisboa.pt (P.M.); 8ISAMB-Environmental Health Institute, Faculty of Medicine, University of Lisbon, 1649-020 Lisbon, Portugal; 9KinesioLab, Research Unit in Human Movement Analysis, Piaget Institute, 2805-059 Almada, Portugal; 10Department of Psychology, University of Geneva, 1205 Geneva, Switzerland; 11Swiss National Centre of Competence in Research LIVES—Overcoming Vulnerability: Life Course Perspectives, 1015 Lausanne, Switzerland

**Keywords:** gait, cognition, aging, physical functions

## Abstract

**Simple Summary:**

Age and mobility are interrelated. In this context, cognitive performance (CP) and physical functions (PF) play a mediating role. However, these concepts are multifaceted, and their interrelationships need further investigations. Thus, our study aims (1) to investigate the association between CP and PF with GS and (2) to examine whether CP and PF mediate the association between age and GS in a large sample of older Brazilian adults. The findings show that low levels of CP and PF were associated with a greater chance of the older individual presenting a slow GS. Moreover, the mediation model indicated that CP and PF mediated, by approximately 12% and 98%, respectively, the association between age and GS.

**Abstract:**

Introduction: With vulnerable aging, gait speed (GS) undergoes progressive changes, becoming slower. In this process, cognitive performance (CP) and physical function (PF) both play an important role. This study aims (1) to investigate the association between CP and PF with GS and (2) to examine whether CP and PF mediate the association between age and GS in a large sample of Brazilian older adults. Methods: A cross-sectional study analyzed 697 individuals (mean age 70.35 ± 6.86 years) from the state of Amazonas. The CP was evaluated by the COGTEL test battery, PF by the Senior Fitness Test battery, and GS with the 50-foot Walk Test. Results: Older adults with a lower CP and PF had a 70% and 86% chance of slow GS, respectively. When CP and PF were placed simultaneously as mediators, the direct effect estimated by the model revealed a non-significant relationship between age and GS. Specifically, CP and PF mediated the association between age and GS, at approximately 12% and 98%, respectively. Conclusions: CP and PF show the potential to estimate GS performance among older adults. Moreover, CP and PF indicated a negative and direct association between age and slow GS, especially PF.

## 1. Introduction

Age-related adaptations during gait lead the older person to adopt a more stable gait pattern [1], which means being more cautious during walking [2]. Thus, in older adults, typical changes in gait include reduced speed [3,4], shorter stride length [5], increased stride time [6], and higher gait stability ratio [7]. Evidence has shown that with vulnerable aging, changes in mobility are strongly associated with poor cognitive performance (CP) [8,9] and reduced levels of physical functions (PF) [10]. This demonstrates that gait, cognition, and PF are interrelated systems [11,12,13].

Gait is a functional activity interrelated with different psychomotor variables that, in turn, contribute to or impair their speed [14]. Among them, there is the motor control [15], muscular performance and musculoskeletal condition [16], and sensory and perceptual function [17]. Losses in these functions occur naturally with increasing age and more rapidly in combination with sedentary habits [18]. A slow GS can induce a vicious cycle of physical deconditioning directly affecting health factors, such as circulatory systems, lungs, and the nervous and musculoskeletal systems [19]. Previous studies have also linked older adult GS with survival and mortality [20,21,22]. In a longitudinal study that examined the relationship between varying measures of time of PF and survival in men and women aged >70 years [23], a 0.1 m/s higher GS performance was seen among women with a 12% reduction in the probability of death in the subsequent year. In contrast, among men, a 0.1% m/s increase in GS represented a reduction in death by 3%. Another point to consider is that the combination of vulnerable aging and low PF levels not only constitutes a vicious cycle harmful to GS, but is also associated with an increased risk of falls [24]. In old age, falls are usually accompanied by fractures and hospitalizations and can even lead to death [25]. Thus, this suggests that GS is a determining factor in maintaining older adults’ health and quality of life [26,27,28].

Another reason to deepen the understanding of GS in vulnerable aging is that its measurement serves as a biomarker of cognitive decline [29,30,31]. Considering the predictive power of gait alteration for incident cognitive decline (a pre-dementia state) [32], Verghese et al. [33] proposed a clinical tool to target interventions for older adults without disabilities or dementia at increased risk of cognitive decline. The tool was titled Motor Cognitive Risk Syndrome (MCR), which compared the diagnosis of mild cognitive impairment (MCI). MCI does not depend on a formal neuropsychological assessment as it is performed solely based on slow gait or cognitive complaints. This strategy is advantageous for being low cost, not requiring language assessment, and/or accounting for years of education. Thus, in recent decades, studies have used the dual-task paradigm during gait to assess the association between slow gait in aging and cognitive deficits [12,34,35,36]. As a complex motor activity, gait requires precise commands from the central nervous system [37]. Thus, a performance <0.80 m/s may reveal functional and/or structural changes in the brain [38], especially in regions associated with the front-subcortical circuit, responsible for mobility and balance [39]. Slow gait may also point to a possible reduction in grey matter volume in the prefrontal cortex [40], reduction in periventricular and subcortical white matter [41], and reduction in hippocampal volumes [31], as well as raising suspicions about possible atrophy of the medial temporal areas [42].

Although changes in GS caused by aging are a recurring theme in the specialized literature [9,11,39], including their associations with CP and PF levels [3,43], the interrelationships, as well as the importance of the different factors involved, remain much less understood [13,17]. Thus, a mediation study can expand the understanding of the mechanisms involved in reduced GS in aging. Second, based on the findings, it will be possible to plan and create more precise therapeutic interventions to prevent the decline in mobility in the older population, consequently benefiting their levels of independence and well-being. Third, understanding the relationship between age, CP, PF, and GS can provide new insights into the area of fall prevention [30]. Finally, the literature shows a lack of studies that include large samples of cognitively healthy older adults, as well as older adults residing in South American countries [44]. Thus, our aim is (1) to investigate the association between CP and PF with GS and (2) to examine whether CP and PF mediate aging-associated decline in GS in a large sample of Brazilian older adults.

## 2. Materials and Methods

### 2.1. Design and Participants

A cross-sectional analytical observational study was carried out with data from the research project “Health, Lifestyle, and Functional Fitness in the Older People from Amazonas, Brazil” (SEVAAI). Data were collected between 2016 and 2017 in the state of Amazonas (municipalities of Manaus, Fonte Boa, and Apuí), located in the northern region of Brazil. The investigation followed the ethical principles contained in Resolution 466/12 of the National Health Council of the Ministry of Health, evaluated and approved by the Ethics Committee in Research with Human Beings of the Universidade do Estado do Amazonas (nº 1.599.258—CAAE: 56519616. 0000.5016). Participants were recruited through newspapers, churches, support centers, groups, or associations of older people in the municipalities of Manaus, Fonte Boa, and Apuí. All participants were informed about the procedures and voluntarily signed an informed consent form before the assessments. The inclusion criteria were: residing in one of the three cities mentioned, minimum age of 60 years, being able to walk independently and performing physical assessments, presenting autonomy and independence to perform activities of daily living, and not indicating serious problems with health (contraindications for exercise). A score of <14/30 in the Mini-Mental State Examination (MMSE) [45] was considered an exclusion criterion (the MMSE was administered to potential participants after they gave consent). Of 701 people who met the search criteria and were included in the original study, 697 were included in the present eligibility:;one was excluded due to Parkinson’s disease and three due to Alzheimer’s disease.

### 2.2. Data Collection

#### 2.2.1. Demographics and Clinical Data

Information on sex, age, years of schooling, falls, and Parkinson’s and Alzheimer’s diseases were collected by self-report, obtained individually in a face-to-face interview, using a health questionnaire employed in FallProof! Program [46]. The interviews were conducted by trained field staff members.

#### 2.2.2. Cognitive Assessment

CP was assessed using the Cognitive Telephone Screening Instrument (COGTEL) test battery [47]. The test is composed of six subtests, which represent important domains of cognitive function: (1) prospective memory; (2) short-term verbal memory; (3) working memory; (4) inductive reasoning; (5) verbal fluency; and (6) long-term verbal memory. The calculation of the total score (continuous scale) was performed using the following formula: COGTEL total score = 7.2 × prospective memory + 1.0 × short-term verbal memory + 0.9 × long-term verbal memory + 0.8 × working memory + 0.2 × verbal fluency + 1.7 × inductive reasoning. Its psychometric properties were verified in the older Brazilian population [48], presenting excellent reliability and high validity. The assessments were conducted individually in face-to-face interviews by specially trained research team members.

#### 2.2.3. Physical Functions

PF was evaluated using the Senior Fitness Test (SFT) battery [49]. For the present study, six PF indicators were selected as physical fitness parameters: (1) Lower limb strength: Participants were asked to get up from a chair, after a signal, and then return to a fully seated position, repeating this action as many times as possible for 30 s. (2) Arm curl, to assess upper body strength: After a signal, participants were instructed to flex and extend the elbow (dominant hand) through its full range of motion, lifting a weight (2.3 kg dumbbell for women and 3.6 kg dumbbell for men) as many times as possible for 30 sec. The total number of repetitions performed was used as a score. (3) Lower body flexibility (sit-and-reach chair/cm): Participants were asked to sit on the edge of a chair, with one leg bent and the other leg extended straight in front, keeping the heel on the floor, without bending the knee. The task consisted of extending the hands forward towards the feet, slowly sliding over the extended leg. The score was determined by the number of centimeters reached beyond the toes (highest score) or reached before the toes (lowest score). (4) Upper body flexibility (back scratching/cm): Participants were asked to place one hand behind the side of the same shoulder with the forearm pronated, the other hand behind the back, and fingers extended. The score was obtained by the centimeters needed for the middle finger to touch the fingers of the other hand (lowest score) or by the centimeters that the middle finger overlapped with the other hand (highest score). (5) Agility/dynamic balance (8 feet up-and-go/s): Participants were asked to sit in a chair, place hands on thighs and feet flat on the floor. After a signal, they got up from the chair, walked as quickly as possible (without running) around a cone placed 8 ft (2.44 m) in front of the chair, returning and sitting fully in the chair. The test result was mensured by the time in seconds used to get up from the sitting position, walk and return to the sitting position. (6) Aerobic endurance (6MWT): Participants were asked to walk, after a signal, as fast as possible (without running) along a marked path. This action occurred as many times as possible. The test score was established by the distance (meters) covered in the six-minute interval (6 min). The calculation of the continuous global measure of the participants’ PF was determined by summing the scores of all six indicators provided by the SFT battery [49].

#### 2.2.4. Gait

GS was assessed using the 30-foot walk test. Participants were required to walk a distance of 30 feet (9 m) at their preferred speed. For each participant, three measures were collected, and the best performance was considered in the analysis. A full description of the test administration instructions for the 30-foot walk test is reported in Rose [46].

#### 2.2.5. Covariates

Through face-to-face interviews, participants reported sex, age, years of education, and the number of falls in the last 12 months.

### 2.3. Statistical Analysis

Categorical variables are presented as frequencies and percentages, while continuous variables are presented as means and standard deviations. The main characteristics of the participants were compared using the chi-squared test (categorical variables) and the unpaired Student’s *t*-test for independent samples (continuous variables). The composition of groups was determined by a cut-off point established by the overall mean of the participant’s performance on the cognitive test (COGTEL), based on the following equation: COGTEL total score [18.9 points] minus 1 standard deviation. The calculation was based on the literature, indicating the method effectively identified individuals with cognitive vulnerability and/or mild cognitive impairment [50]. Thus, two groups were established: mild cognitive impairment with <17.9 points vs. normal cognition with ≥17.9 points. In a second step, two separate logistic regression analyses were performed to test cross-sectional associations between CP and PF with GS (the study’s first objective). Thus, based on the performance of CP and PF, the chance of older adults presenting a low GS was evaluated. Three models were calculated: Model 1 unadjusted; Model 2 adjusted by sex and age; Model 3 adjusted by sex, age, MMSE, and years of education. The insertion order was from highest to lowest (forward model), respecting the magnitude of Spearman’s correlation coefficient. The odds ratio (OR) and their respective confidence intervals (95% CI) were used to present the results. Effect size estimates were reported using the adjusted R^2^.

Finally, a mediation analysis was performed to examine whether CP and PF mediate the association between age and GS (the second objective of the study). A mediation, or indirect effect, occurs when the causal effect of an independent variable *X* (age) can predict the dependent variable *Y* (GS) transmitted by mediators *M*_1_ and *M*_2_ (CP and PF) [51]. A complete mediation is observed if the joint inclusion of objectively measured *M*_1_ and *M*_2_ reduces the observed association between *X* and *Y* to non-significance. A partial mediation occurs if the observed association between *X* and *Y* becomes weaker after the inclusion of *M*_1_ and *M*_2_. An indirect effect was considered significant when the confidence interval did not include zero. Mediation hypotheses were tested using the bias-corrected bootstrap method with 5000 samples to calculate confidence intervals (95%). To perform the analysis, a computational complement of the SPSS program was applied using PROCESS v4.0, a model estimation analysis developed by Hayes [52]. The significance level for all analyses was defined as *α* < 0.05.

## 3. Results

### 3.1. Main Characteristics of the Participants

Table 1 presents the participants’ characteristics stratified by cognitive status. A total of 697 participants was included in the study. Of these, 48.2% had mild cognitive impairment. The mean age was 70.35 ± 6.86, with 61.7% being female. Except for the falls variable, all the others presented different statistical results (*p* < 0.001). Moreover, members of the group without cognitive deficit indicated better results in most variables (years of education, MMSE, COGTEL, total PF, and GS).

### 3.2. Associations between CP and PF with GS

Table 2 presents the associations and odds ratios between a low level of CP and PF with GS. After adjusting for covariates (e.g., sex, age, MMSE, and years of education), older adults with a low CP indicated a greater chance of having a low GS than those with a high CP (OR = 0.296, 95% CI = 0.128–0.302, *p* < 0.001, R^2^ = 0.301). Moreover, after adjusting for covariates (e.g., sex, age, MMSE, and years of education), older adults with low PF level also indicated a greater chance of having low GS (OR = 0.141, 95% CI = 0.091–0.198, *p* < 0.001, R^2^ = 0.310).

### 3.3. Mediation Analysis: CP and PF in the Relationship between Age and GS

Figure 1 presents the results of the multiple mediation analysis. The total effect of the model (*x*–*y*) showed a significant negative relationship between increasing age and low GS performance (β = −0.011, 95% CI = −0.0164–0.0064, *t* = −4.4934, *p* < 0.001). Model 1 was controlled for confounders (i.e., sex and years of education) and showed that age (independent variable) had a negative and significant association with the CP mediator (β = −0.254, 95% CI = −0.3552–0.1538, *t* = −4.9611, *p* < 0.001), and also with the PF mediator (β = −4.6549, 95% CI = −5.6386–3.6712, *t* = −9.2909, *p* < 0.001). Model 2 shows significant and positive associations between the CP mediator (β = 0.023, 95% CI = 0.0203–0.0265, *t* = 14.7091, *p* < 0.001) and the PF mediator (β = 0.001, 95% CI = 0.0008–0.0015, *t* = 7.0577, *p* < 0.001) with GS (dependent variable). When the mediating variables (m1 and m2) were included, the direct effect estimated by the model (*x*–*y*) revealed a non-significant relationship between age and GS (β = −0.001, 95% CI = −0.0047–0.0044), *t* = −0.0591, *p* = 0.9529). The indirect effects showed that both CP (β = −0.006, 95% CI BCa = −0.0085–0.0035) and PF (β = −0.005, 95% CI BCa = −0.0074–0.0036) were independent mediators of the negative effect that aging has on the GS performance of older adults. Thus, the proportion of the total effect of age on GS mediated by CP and PF was approximately 12% and 98%, respectively.

## 4. Discussion

The present study aimed to investigate the association between CP and PF with GS and to examine whether CP and PF mediate the association between age and GS in a large sample of Brazilian older adults. Our main findings were the following: First, based on the results of the logistic regression, in the final model controlling for sex, age, MMSE, and years of education, older adults with a lower CP and lower PF levels indicated a chance of having a slow GS, in 70% and 86%, respectively. Second, when CP and PF were placed simultaneously as mediators, the observed association between *x* and *y* became weaker, and the direct effect estimated by the model revealed a non-significant relationship between age and GS. Thus, CP and PF were able to partially mediate the association between age and GS in approximately 12% and 98%, respectively.

To the best of our knowledge, this is the first study to estimate the mediation exerted by CP and PF in the association between age and GS in a large sample of Brazilian older adults. Thus, our findings may contribute to the current knowledge, providing in-depth evidence on the role that CP and PF play in the relationship between age and GS in the older population. Overall, our findings are in agreement with previous investigations; PA levels may be a possible strategy to improve GS performance over aging [3]. Logically, the second point also depends on the local creation of health, infrastructure, and education policies that allow the older population to practice physical exercises [53]. As a result, we can expect an improvement in CP and PF levels [54]. In the case of the older population, a medium to high level of PA is positively associated with a good physical performance [55]. The measure is recommended as a strategy to reduce health risks [56], such as cardiovascular diseases and diabetes, as well as a preventive factor for falls [57], which are strongly associated with a slow gait [58], and consequently increased risk of mortality [59]. It is worth noting that, after another fall [60], older adults tend to adopt a slower gait due to fear of another fall [61,62]. Moreover, falls generate high costs for health services, being considered a public health problem [63,64].

The associations of the mediation analysis revealed by the present study may contribute to a conceptual advance on the role of age in GS, as they demonstrated, in percentage terms, how CP and PF act in the relationship exerted by aging on GS. The findings also showed that, in our sample, the percentage of mediation indicated by CP was approximately eight times lower than the value indicated by PF. So, the CP may have a direct impact on PF. Therefore, a practical implication is to increase PF, which in turn will benefit GS. Among age-related changes, the reduction in muscle strength resulting from the loss of skeletal muscle mass (sarcopenia) and is one of the limiting agents for older adults to present adequate PF performance [64]. Changes in the musculoskeletal system directly affect spatial and temporal parameters of the gait, including speed, cadence, and stride length [65]. The literature highlights that a longer stride length during gait benefits biomechanical efficiency [66]. On the other hand, an increase in stride-time variability may be associated with abnormally high cortical levels of gait control, altering the stride mechanism [65]. This confirms the mediating role that PF and CP have in the relationship between age and GS. A longitudinal study [67] and systematic review with a meta-analysis [11] showed a causal relationship between poor performance on measures of PF (including gait parameters) and the worsening of CP. In a population-based study (n = 3460; ≥65 years), an association was shown between CP and a reduced probability of disability in several domains of PF [68]. It is worth noting that low performance on the 6MWT aerobic endurance test not only suggests possible heart failure, as it also reflects a low GS [69]. Therefore, short steps, in addition to reducing GS, serve as a prognosis of heart failure [66]. Although our study did not focus on age-related changes in body composition, it is important to note that there is strong evidence of an association between BMI in the obesity range and physical disability in older adults [70]. A relationship between older adult obesity and cognitive impairment has been indicated by population-based studies [71,72]. Therefore, a cohort study (n = 572; ≥60 years) carried out to assess the relationship between anthropometric measurements and PF showed greater associations between muscle mass index and anthropometric measures of central fat with worse PF performance for both sexes [73]. In this context, an effective strategy capable of reducing the physical frailty of the obese older adult population focuses on increasing levels of physical activity associated with a healthy lifestyle [74].

As for the study’s strengths, we can mention the analysis based on a large sample of older adults. Second, the three municipalities where the study was carried out are located in the northern region of the Brazilian territory, which, compared to the southern and southeastern regions of Brazil, is considered a territory of social and economic vulnerability [75,76]. This means that the present study brought to light, for the first time, information on aspects of GS, CP, and PF of community-dwelling older adults in Brazil, who age in precarious conditions [76]. Third, the instruments used to assess CP, PF, and GS are reliable and valid measures [46,47,48,49]. Moreover, the novelty of examining the mediating role of CP and PF in the association between age and GS of older adult Brazilians is a further strength. Thus, our findings can inform public policies to improve the quality of life and healthy aging of older adult Brazilians [77], especially those in vulnerable situations [78]. Although the present study did not intend to deepen comparisons by sex, it is worth noting that the descriptive analysis showed that, comparatively, women had proportionally greater cognitive deficits than men. Moreover, older adults in the group with cognitive impairment had four times fewer years of education than those in the group without cognitive impairment. Regarding study limitations, we highlight the cross-sectional design, limiting causal conclusions. Therefore, the present study encourages future investigations based on longitudinal approaches. A further possible focus for future research would be to investigate the age-related decline in GS, mediated by CP and PF, including sex as a potential moderator, to determine whether there is a difference between older men and women. Additionally, as the PF reveals a greater role as a mediator of the association between age and GS, it will be interesting to investigate the role of PF as a mediator of CP and gait speed.

## 5. Conclusions

The findings revealed in this large sample of older adults from three municipalities in the northern region of Brazil highlighted the role that CP and PF have in the relationship between age and GS performance. Moreover, CP and PF explained the negative association between age and slow GS. These results reinforce the importance of older adults adopting an active lifestyle as a possible strategy for maintaining PF and thereby also adequate GS levels. Our results also strengthen the essential role that a preserved cognitive function during aging offers for GS, which in turn is a determinant of motor capacity for older adults’ autonomy. Finally, based on the findings, local health policies and interventions can be planned and/or (re)directed to promote active [79] and successful aging [80] in the northern region of Brazil.

## Figures and Tables

**Figure 1 geriatrics-07-00073-f001:**
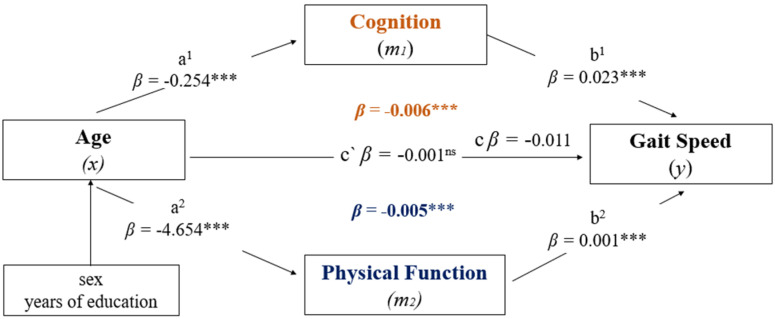
Mediation analysis: CP and PF in the relationship between age and GS. Parallel mediation analysis of the effects of age (aging) on GS (m·s^−1^) through CP (COGTEL, total score) and PF (Senior Fitness Test, total score), adjusted for sex and years of education. The analysis considered the number of bootstrap samples = 5000. The indirect effect was statistically significant at the 95% confidence interval (CI) when the CI did not include 0. Betas (β) are reported as the product of simultaneous regression with substitution of bootstrap: (1) Path a^1^ and a^2^ = association between age and CP and PF, respectively, and (2) Path b^1^ and b^2^ = association between CP and PF with GS; (3) Path c’ = direct effect (*x*–*y*): *m*_1_(brown color) and *m*_2_ (blue color) associations = indirect effect (*x*–*y*) by CP and PF, respectively. NS = not significant; *** *p* < 0.001, c = total effect; c’ = direct effect; a = path Model 1; b = path Model 2.

**Table 1 geriatrics-07-00073-t001:** Main characteristics of the sample.

Variable	Full Sample (n = 697)	Cognitive Impairment (n = 336)	No Cognitive Impairment (n = 361)	*p*-Value
Age (years)	70.35 ± 6.86	71.58 ± 7.47	69.23 ± 6.05	<0.001
Sex n (%)				0.001
Female	430 (61.7)	183 (55.3)	247 (67.5)	
Male	267 (38.3)	148 (44.7)	119 (32.5)	
Education (years) (n)	5.35 ± 5.54	1.97 ± 2.90	8.40 ± 5.61	<0.001
Falls (n)	0.59 ± 1.32	0.68 ± 1.66	0.51 ± 0.90	0.087
MMSE (n)	24.40 ± 4.23	22.12 ± 4.16	26.51 ± 3.03	<0.001
Cognition (n)				
COGTEL (score)	18.95 ± 9.45	11.11 ± 4.59	26.17 ± 6.59	<0.001
Physical function (n)				
PF total (score)	470.63 ± 96.16	461.09 ± 99.95	479.40 ± 91.81	<0.001
Gait (m/s)				
GS	1.35 ± 0.47	1.12 ± 0.34	1.55 ± 0.48	<0.001

Note: MMSE: Mini-mental state examination; PF: physical functions; GS: gait speed; m/s: meters per second. *p* ≤ 0.005.

**Table 2 geriatrics-07-00073-t002:** Association between physical functions and cognitive performance.

Variable	Model 1 OR 95% CI *p*-Value	Model 2 OR 95% CI *p*-Value	Model 3 OR 95% CI *p*-Value
Lower CP	0.450 (0.359–0.484) <0.001	0.437 (0.347–0.473) <0.001	0.296 (0.128–0.302) <0.001
Lower PF	0.206 (0.125–0.261) <0.001	0.188 (0.105–0.247) <0.001	0.141 (0.091–0.198) <0.001

Note: CP: cognitive performance; PF: physical functions; Model 1: unadjusted; Model 2: adjusted by sex and age; Model 3: adjusted by sex, age, MMSE, and years of education. *p* < 0.001.

## Data Availability

The data presented in this study are available upon request from the corresponding author.

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
