# Peer review of "The Role of Cognitive Performance and Physical Functions in the Association between Age and Gait Speed: A Mediation Study"

_geriatrics, 2022, doi:10.3390/geriatrics7040073_

Round 1
Reviewer 1 Report
This is an interesting study in relation to healthy ageing, and in particular, gait speed. The USP of the paper is that it explores mediating role of cognitive and function in terms of the association between age and gait speed South American older adults.
Abstract
Line 29: Be a little more specific on the changes in gait speed.
Line 30: “… and physical function (PF) both play…”
Introduction
Lines 55-56: Update sentence to: “Thus, in older adults, typical changes in gait include reduced speed [3,4], shorter stride length [5], increased stride time [6], and higher gait stability ratio [7].”
Line 59: Remember to abbreviate physical function to PF.
Line 75: Replace “the facts suggest” with “this suggests”
Line 82: MCI abbreviation must be fully highlighted.
Materials and Methods
Line 125: It needs to be made clearer when the MSSE was administered to potential participants, before or directly after they gave consent? Also, it is unclear why you selected 14/30 from the referenced Cochrane review? From my understanding of the MMSE, higher scores suggest more normal cognitive function. However, you have excluded these individuals. I thought this study was focusing cognitively healthy older adults? Please clarify these issues.
Lines 128-129: How was this determined? Self-reported by participants or through medical records?
Line 134: Highlight the COGTEL abbreviation fully before shortening.
Lines 147-152: Add a little more detail for how each test was administered (e.g. was test 2 conducted standing or sitting using the dominant arm?, length of test 6 course etc).
Lines 152-153: It should be made clearer the thresholds for how each test is scored as well as inclusion of a suitable reference regarding the scoring.
Line 155: I can only see one gait speed (assuming preferred gait speed) being reported in the Results. The way this part of the Methods is worded suggests you collected both maximum and preferred values…? Please clarify.
Lines 180-181: Did you not collinearity between years of education and MSSE in your regression models? I would have thought these could be highly correlated?
Lines 182-183: Consider including a reference to justify use of p <0.2 for inclusion of control factors.
Line 199: Should be less than 0.05 (<0.05).
Results
Table 1: The years of education for those with cognitive impairment is less than 2 years. This seems incredibly low!
Line 206: Not strictly true, there was no significant difference in falls risk between the groups.
Lines 217-220: The written 95% CIs for CP and PF do not fully match the ones in Table 2. Please update this accordingly.
Table 2: It seems unusual to me that the odds ratio for “lower PF” rises from Model 2 to Model 3 after full adjustment, considering the pattern for “lower CP”. It might be worth checking your models again.
Lines 238-241: A couple of things. Replace “CI 95%” with “95% CI”. Also, for CP, is the upper CI -0.0035 or +0.0035? I am assuming it should be written as “= -0.0085 – -0.0035” but it is currently not clear. For PF, is this actually significant? The 95% CI is “-0.0074 – 0.0036” which would suggest it is not? This could be my ignorance so apologies if so but please clarify this. If the above 95% CI is correct, does this then not suggest this is insignificant?
Figure 2: Replace “Physycal Function” with “Physical Function”.
Discussion
Line 260: Should it not be 70% and 77%?
Line 264: It is not clear where the 12% and 98% figures come from. Unless I have missed it, I can see no reference to this in the Results section. Please update accordingly.
Lines 269-270: Your study has not focused on exercise and physical activity levels so not sure you can make this statement. Either delete or revise accordingly.
Line 288: Should be “PF”, not “FP”. This occurs in a few places in the Discussion section so please update accordingly.
Line 302: What about generalisability issues such as education levels and sex distribution?
Author Response
Reviewer 1
Dear Reviewer, we are grateful for all the comments, and are available for future clarifications and/or corrections.
* Changes were made in the text using Microsoft Word's built-in track changes function.
Abstract
- Line 29: Be a little more specific on the changes in gait speed.
Reply
Dear Reviewer, the sentence has been supplemented in the Abstract
- Line 30: “… and physical function (PF) both play…”
Reply
Dear Reviewer, the sentence has been corrected in the Abstract.
Introduction
- Lines 55-56: Update sentence to: “Thus, in older adults, typical changes in gait include reduced speed [3,4], shorter stride length [5], increased stride time [6], and higher gait stability ratio [7].”
Reply
Dear Reviewer, the sentence has been corrected (page 2, line 54-56)
- Line 59: Remember to abbreviate physical function to PF.
Reply
Dear Reviewer, this abbreviation has been adjusted throughout the text.
- Line 75: Replace “the facts suggest” with “this suggests”
Reply
Dear Reviewer, the sentence has been adjusted (page 2, from line 75-76)
- Line 82: MCI abbreviation must be fully highlighted.
Reply
Dear Reviewer, the sentence has been adjusted (page 2, from line 82-83)
Materials and Methods
- Line 125: It needs to be made clearer when the MSSE was administered to potential participants, before or directly after they gave consent? Also, it is unclear why you selected 14/30 from the referenced Cochrane review? From my understanding of the MMSE, higher scores suggest more normal cognitive function. However, you have excluded these individuals. I thought this study was focusing cognitively healthy older adults? Please clarify these issues.
Reply
Dear Reviewer, we clarified this information and corrected the typo: “A score of < 14/30 in the Mini-Mental State Examination (MMSE) [45] was considered an exclusion criterion (the MMSE was administered to potential participants after they gave consent).”
- Lines 128-129: How was this determined? Self-reported by participants or through medical records?
Reply
Dear Reviewer, Information on data collection regarding Parkinson's and Alzheimer's disease has been included in the text (page 3, line 133-137)
- Line 134: Highlight the COGTEL abbreviation fully before shortening.
Reply
Dear Reviewer, Cognitive Telephone Screening Instrument (COGTEL) test name has been detailed (page 3, 141)
- Lines 147-152: Add a little more detail for how each test was administered (e.g. was test 2 conducted standing or sitting using the dominant arm?, length of test 6 course etc).
Reply
Dear Reviewer, all STF battery tests have been better detailed (page 3, starting at line 153)
- Lines 152-153: It should be made clearer the thresholds for how each test is scored as well as inclusion of a suitable reference regarding the scoring.
Reply
Dear Reviewer, STF battery tests have been detailed, plus a reference on application and scores has been included (page 4, line 181)
- Line 155: I can only see one gait speed (assuming preferred gait speed) being reported in the Results. The way this part of the Methods is worded suggests you collected both maximum and preferred values…? Please clarify.
Reply
Dear Reviewer, the sentence has been corrected. In the present study, only the preferred gait speed was used (page 4, line 185)
- Lines 180-181: Did you not collinearity between years of education and MSSE in your regression models? I would have thought these could be highly correlated?
Reply
Dear Reviewer, to clarify, there was no collinearity in the models. Besides that, note that MMSE and education were only considered as control variables that were not the main variables of interest in our study.
- Lines 209: Consider including a reference to justify use of p <0.2 for inclusion of control factors.
Reply
Dear Reviewer, to clarify, this approach allows the compromise of identifying and including the most important control variables, while minimizing the bias of overfitting by introducing an excessively large number of control variables. Only the most important were included in the models, starting with those that show the greatest associations. The insertion order was from highest to lowest (forward model), respecting the magnitude of Spearman’s correlation coefficient. This issue is explained in the Statistical Analysis section.
- Line 199: Should be less than 0.05 (<0.05).
Reply
Dear Reviewer, the sentence has been corrected (page 5, line 226).
Results
- Table 1: The years of education for those with cognitive impairment is less than 2 years. This seems incredibly low!
Reply
Dear Reviewer, we have reviewed the database and proceeded with a new statistical analysis for years of education, and we have confirmed these values. Really, they are low. This is due to the conditions of extreme socioeconomic vulnerability of the population residing in the Northern region of Brazil.
- Line 206: Not strictly true, there was no significant difference in falls risk between the groups.
Reply
Dear Reviewer, we reviewed the database and also performed a new statistical analysis for falls, and we confirmed that there was no different statistical result.
- Lines 217-220: The written 95% CIs for CP and PF do not fully match the ones in Table 2. Please update this accordingly.
Reply
Dear Reviewer, the values of 95% CIs for CP and PF have been corrected, as per Table 2 (line 244-245)
- Table 2: It seems unusual to me that the odds ratio for “lower PF” rises from Model 2 to Model 3 after full adjustment, considering the pattern for “lower CP”. It might be worth checking your models again.
Reply
Dear Reviewer, we have reviewed the odds ratio for “lower PF” from Model 3 (Table 2; page 6, line 246). And we also corrected this percentage in Abstract.
- Lines 238-241: A couple of things.
Reply
Dear Reviewer, we corrected the sentence.
- Replace “CI 95%” with “95% CI”;
Reply
Dear Reviewer, the sentence has been revised (page 6, line 267).
- Also, for CP, is the upper CI -0.0035 or +0.0035? I am assuming it should be written as “= -0.0085 – -0.0035” but it is currently not clear;
Reply
Dear Reviewer, we have reviewed all values and their respective signs ( –/ - ; long and short) in section 3.3 Mediation analysis (page 6, starting at line 253)
- For PF, is this actually significant? The 95% CI is “-0.0074 – 0.0036” which would suggest it is not? This could be my ignorance so apologies if so but please clarify this. If the above 95% CI is correct, does this then not suggest this is insignificant?
Reply
Dear Reviewer, we have reviewed all values and their respective signs ( –/ - ; long and short) in section 3.3 Mediation analysis (page 6, starting at line 253). Both CI boundaries are negative (-0.0074 – -0.0036), thus it is significant.
- Figure 2: Replace “Physycal Function” with “Physical Function”.
Reply
Dear Reviewer, the Figure has been corrected and replaced.
Discussion
- Line 260: Should it not be 70% and 77%?
Reply
Dear Reviewer, we agree with your suggestion. A percentage has been corrected/rounded in the Discussion section (page 7, line 287) as well as in the Summary. The other percentage was corrected, due to the revision of the Model 3 result.
- Line 264: It is not clear where the 12% and 98% figures come from. Unless I have missed it, I can see no reference to this in the Results section. Please update accordingly.
Reply
Dear Reviewer, we have included the respective 12% and 98% results at the end of section 3.3 Mediation analysis (page 6, lines 268-269).
- Lines 269-270: Your study has not focused on exercise and physical activity levels so not sure you can make this statement. Either delete or revise accordingly.
Reply
Dear Reviewer, we agree that the sentence should be removed.
- Line 288: Should be “PF”, not “FP”. This occurs in a few places in the Discussion section so please update accordingly.
Reply
Dear Reviewer, we have corrected this acronym throughout the text
- Line 302: What about generalisability issues such as education levels and sex distribution?
Reply
Dear Reviewer, information about sex and years of education has been entered (page 7, lines 302-306)
Reviewer 2 Report
It is a very interesting paper, because physical function play an important role for older people and it is well evidence based. This paper has good flow, however a few amendments are suggested.
Introduction:
Please insert the role of physical activity and exercise in the physical function of older people
What is the primary and secondary aims? What about hypothesis?
What are the gaps which leads to your aims?
Methods:
Sample: Add more. How were recruited
Assessment. Why did you choose this evaluation (Reliability, validity of the assessment tools)
Discussion:
What is the expected PF for this population?
What new has been added by this research? Clinical implications.
Any limitations?
Additional references for Intro and discussion (i.e. Line 290)
Effects of multicomponent exercise training intervention on hemodynamic and physical function in older residents of long-term care facilities: A multicenter randomized clinical controlled trial. Journal of Bodywork and Movement Therapies. 2021; 28: 231-237. https://doi.org/10.1016/j.jbmt.2021.07.009.
Chase J-AD. Interventions to Increase Physical Activity Among Older Adults: A Meta-Analysis. The Gerontologist. 2015;55(4): 706–718. https://doi.org/10.1093/geront/gnu090.
Pepera G. Randomized Controlled Trial of Group Exercise Intervention for Fall Risk factors Reduction in nursing home residents. Canadian Journal on Aging, 42 (1).
Izquierdo M, Duque G, Morley JE. Physical activity guidelines for older people: knowledge gaps and future directions. The Lancet Healthy Longevity. 2021;2(6): e380–e383. https://doi.org/10.1016/S2666-7568(21)00079-9
Author Response
Reviewer 2
Dear Reviewer, we are grateful for all the comments, and are available for future clarifications and/or corrections.
* Changes were made in the text using Microsoft Word's built-in track changes function.
It is a very interesting paper, because physical function play an important role for older people and it is well evidence based. This paper has good flow, however a few amendments are suggested.
Reply
Dear Reviewer, thank you for this overall positive evaluation. We have addressed all concerns. Please see our detailed responses below.
Introduction:
- Please insert the role of physical activity and exercise in the physical function of older people
Reply
Dear Reviewer, we agree that expanding on these issues might be interesting. However, in response to Reviewer 1, we had to delete all notions regarding physical activity and exercise since he/she argued that our study does not focus on exercise and physical activity.
- What is the primary and secondary aims? What about hypothesis?
Reply
Dear Reviewer, in lines 105-107 our objectives were described: "Thus, our aim was (1) to investigate the association between CP and PF with GS and (2) to examine whether CP and PF mediate aging-associated decline in GS in a large sample of Brazilian older adults". Due to the lack of research on the special population targeted (extreme socioeconomic vulnerability in Northern Brazil), we did not formulate specific hypotheses.
- What are the gaps which leads to your aims?
Reply
Dear Reviewer, we have clarified the gaps that motivated our study in more detail.
Methods:
- Sample: Add more. How were recruited
Reply
Dear Reviewer, the information has been included in the text (page 3, line 118-120).
- Assessment. Why did you choose this evaluation (Reliability, validity of the assessment tools)
Reply
Dear Reviewer, we clarify that “the instruments used to assess CP, PF, and GS are reliable and valid measures [46,47,48,49].”
Discussion:
- What is the expected PF for this population?
Reply
Dear Reviewer, due to the lack of research on this special population targeted (extreme socioeconomic vulnerability in Northern Brazil), we did not have specific a priori expectations regarding their exact PF level.
- What new has been added by this research? Clinical implications.
Reply
Dear Reviewer, our study allows the following conclusions: “The findings revealed in this large sample of older adults from three municipalities in the northern region of Brazil highlighted the role that CP and PF have in the relationship between age and GS performance. Moreover, CP and PF explained the negative association between age and slow GS. These results reinforce the importance of older adults adopting an active lifestyle as a possible strategy for maintaining PF and thereby also adequate GS levels. Our results also strengthen the essential role that a preserved cognitive function during aging offers for GS, which in turn is a determinant of motor capacity for older adults’ autonomy. Finally, based on the findings, local health policies and interventions can be planned and/or (re)directed to promote active [71] and successful aging [72] in the northern region of Brazil.”
- Any limitations?
Reply
Dear Reviewer, we discuss the study limitations in more detail: “Regarding study limitations, we highlight the cross-sectional design limiting causal conclusions. Therefore, the present study encourages future investigations based on longitudinal approaches. A further possible focus for future research would be to investigate the age-related decline in GS, mediated by CP and PF, including sex as a potential moderator, to determine whether there is a difference between older men and women. Also, as the PF reveals a greater role as a mediator of the association between age and GS, it will be interesting to investigate the role of PF as a mediator of CP and gait speed.”
- Additional references for Intro and discussion (i.e. Line 290)
Reply
A) Dear Reviewer, in response to Reviewer 1, this sentence has been changed.
B) Dear Reviewer, thank you for your suggestions (three articles). However, in response to Reviewer 1, we had to exclude all notions about physical activity and exercise as he argued that our study does not focus on exercise and physical activity.
Round 2
Reviewer 1 Report
Thank you for addressing my comments. There are just a small number of minor aspects to update.
Abstract
Line 30: “functions” should not be plural; should be “function”.
Materials and Methods
Lines 130: Replace “search” with “eligibility”.
Results
Line 234-235: Sorry, my original query could have been clearer. You say “Members of the group without cognitive impairment indicated better results on all variables…”. However, as you highlight, there was no significant between-group differences regarding. You just need to revise the sentence to reflect that MOST variables were better for the group without cognitive impairment apart from falls where there was no significant difference.
Author Response
Dear Reviewer, we thank you for your careful reading and analysis of our manuscript. Your suggestions and requests have been worked out in the text by Microsoft Word's built-in track changes function. If necessary, we are available for future adjustments.
*We inform you that in addition to your suggestions, the text contains changes from Reviewer 2, mainly in the Discussion section.
Summary
Line 30: “functions” must not be plural; should be "function".
Reply
Dear Reviewer, the word has been corrected
Materials and methods
Lines 130: Replace “survey” with “eligibility”.
Reply
Dear Reviewer, the word has been corrected
Results
Line 234-235: Sorry, my original query could have been clearer. You say “Members of the group without cognitive impairment indicated better results on all variables…”. However, as you highlight, there was no significant between-group differences regarding. You just need to revise the sentence to reflect that MOST variables were better for the group without cognitive impairment apart from falls where there was no significant difference.
Reply
Dear Reviewer, Thank you for your remark. The sentence has been corrected (lines 234-237)
Reviewer 2 Report
Dear authors,
The paper was improved and it is a very interesting paper, however, there are still some concerns.
You havent dabated enough your result with previous research in the discussion part.
Say more how PF relates with anthropometric variables and physical fitness. There is evidence on that, which you shouldnt ignore. You havent related enough your results with previous studies.
Moreover, does functional tests i.e. 6MWT (please change to this acronym) performance relates to biomechanical (i.e gait) parameters and how?
Is there any research in any population that could support the relationship between PF (or functional capacity else) with biomechanical parameteres (i.e gait).
Please improve your discussion. Make a better story to connect your results with previous studies
Suggested REFs
Influence of step length on 6-minute walk test performance in patients with chronic heart failure. Physiotherapy 98 (4), 325-329. https://doi.org/10.1016/j.physio.2011.08.005
Association of Neurocognitive and Physical Function With Gait Speed in Midlif. JAMA 2019 2;2(10):e1913123. DOI: 10.1001/jamanetworkopen.2019.13123
Effects of multicomponent exercise training intervention on hemodynamic and physical function in older residents of long-term care facilities: A multicenter randomized clinical trial. Journal of Bodywork and Movement Therapies 28, 231-237. https://doi.org/10.1016/j.jbmt.2021.07.009
Gait, physical function, and physical activity in three groups of home-dwelling older adults with different severity of cognitive impairment – a cross-sectional study. BMC Geriatr 21, 670 (2021). https://doi.org/10.1186/s12877-021-02598-9
Author Response
Dear Reviewer, we thank you for your careful reading and analysis of our manuscript. Your suggestions and requests have been worked out in the text by Microsoft Word's built-in track changes function. If necessary, we are available for future adjustments.
Say more how PF relates with anthropometric variables and physical fitness. There is evidence on that, which you shouldnt ignore. You havent related enough your results with previous studies. Moreover, does functional tests i.e. 6MWT (please change to this acronym) performance relates to biomechanical (i.e gait) parameters and how?
Is there any research in any population that could support the relationship between PF (or functional capacity else) with biomechanical parameteres (i.e gait).
Please improve your discussion. Make a better story to connect your results with previous studies
Reply
Dear Reviewer, the acronym has been corrected (line 179)
Replay
Dear Reviewer, in the Discussion section between lines 318-343, we have inserted information about your considerations, referring to the following topics: (1) biomechanical performance, (2) loss of skeletal muscle mass (sarcopenia) and PF, (3) age-related changes in body composition, (4) the association between poor performance on the 6MWT aerobic endurance test and possible heart failure, and (5) references from population-based studies and a longitudinal study.
* Thanks for the reference list